# A Modern Take on the Bias-Variance Tradeoff in Neural Networks

## Abstract

We revisit the bias-variance tradeoff for neural networks in light of modern empirical findings. The traditional bias-variance tradeoff in machine learning suggests that as model complexity grows, variance increases. Classical bounds in statistical learning theory point to the number of parameters in a model as a measure of model complexity, which means the tradeoff would indicate that variance increases with the size of neural networks. However, we empirically find that variance due to training set sampling is roughly *constant* (with both width and depth) in practice. Variance caused by the non-convexity of the loss landscape is different. We find that it decreases with width and increases with depth, in our setting. We provide theoretical analysis, in a simplified setting inspired by linear models, that is consistent with our empirical findings for width. We view bias-variance as a useful lens to study generalization through and encourage further theoretical explanation from this perspective.

## 1 Introduction

The traditional view in machine learning is that increasingly complex models achieve lower bias at the expense of higher variance. This balance between underfitting (high bias) and overfitting (high variance) is commonly known as the *bias-variance tradeoff* (Figure 1). In their landmark work that initially highlighted this bias-variance dilemma in machine learning, Geman et al. (1992) suggest that larger neural networks suffer from higher variance. Because bias and variance contribute to test set performance (through the bias-variance decomposition), this provided strong intuition for how we think about generalization capabilities of large models. Learning theory supports this intuition, as most classical and current bounds on generalization error grow with the size of the networks (Brutzkus et al., 2018).

However, there is a growing amount of evidence of *larger* networks generalizing *better* than their smaller counterparts in practice (Neyshabur et al., 2014; Novak et al., 2018; Zhang et al., 2017; Canziani et al., 2016). This apparent mismatch between theory and practice is due to the use of worst-case analysis that depends only on the model class, completely agnostic to data distribution and without taking optimization into account.[1] A modern empirical study of bias-variance can take all of this information into account.

We revisit the bias-variance tradeoff in the modern setting, focusing on how variance changes with increasing size of neural networks that are trained with optimizers whose step sizes are tuned with a validation set. In contrast to the traditional view of the bias-variance tradeoff (Geman et al., 1992), we find evidence that the overall variance *decreases* with network width (Figure 1). This can be seen as the "bias-variance analog" of the described recent evidence of larger networks generalizing better. More in line with the tradeoff, we find that variance grows slowly with depth, using current best practices.

To better understand these coarse trends, we develop a new, more fine-grain way to study variance. We separate variance due to initialization (caused by non-convexity of the optimization landscape) from variance due to sampling of the training set. Surprisingly, we find that variance due to training set sampling is roughly *constant* with both width and depth (Figure 2). Variance due to initialization

---

[1] Some recent work has gone in the direction of taking this information into account, see e.g Kuzborskij and Lampert (2018); Dziugaite and Roy (2017).

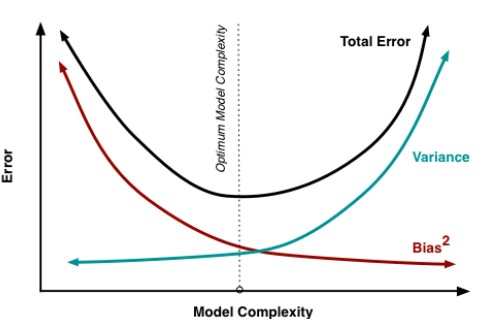 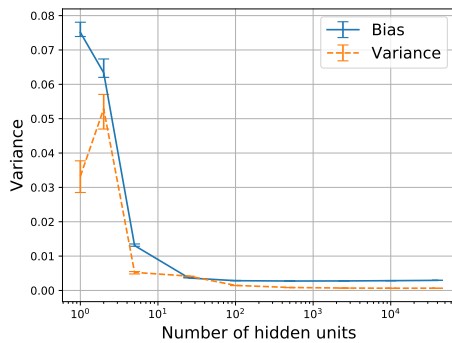

Figure 1: On the left is an illustration of the common intuition for the bias-variance tradeoff (Fortmann-Roe, 2012). We find that variance decreases along with bias when increasing network width (right). These results seem to contradict the traditional intuition.

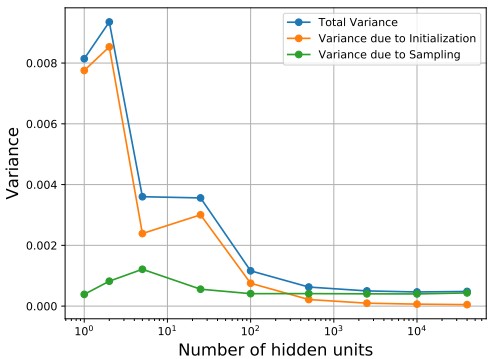 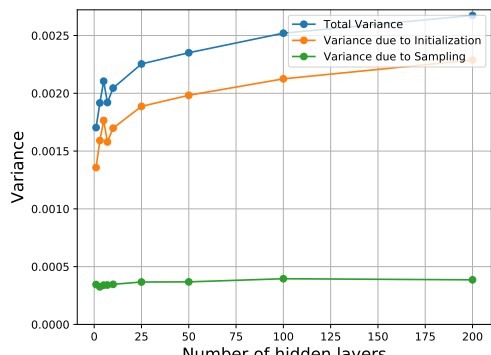

Figure 2: Trends of variance due to sampling and variance due to initialization with width (left) and with depth (right). Variance due to sampling is roughly *constant* with both width and depth, in contrast with what the bias-variance tradeoff might suggest. Variance due to initialization differentiates the effects of width and depth and is in line with neural network optimization literature.

decreases with width and increases with depth, in our setting (Figure 2). To support our empirical findings, we provide a simple theoretical analysis of these sources of variance by taking inspiration from over-parameterized linear models. We see further theoretical treatment of variance as a fruitful direction for better understanding complexity and generalization abilities of neural networks.

MAIN CONTRIBUTIONS

1. We revisit the bias-variance analysis in the modern setting for neural networks and point out that it is not necessarily a tradeoff as overall variance *decreases* with width (similar to bias), yielding *better generalization*.

2. We perform a more fine-grain study of variance in neural networks by decomposing it into variance due to initialization and variance due to sampling. Variance due to sampling is roughly *constant* with both width and depth. Variance due to initialization decreases with width, while it increases with depth, in the settings we consider.

3. In a simplified setting, inspired by linear models, we provide theoretical analysis in support of our empirical findings for network width.

The rest of this paper is organized as follows. Section 2 establishes necessary preliminaries. In Section 3 and Section 4, we study the impact of network width and network depth (respectively) on variance. In Section 5, we present our simple theoretical variance analysis.

## 2 PRELIMINARIES

### 2.1 SET-UP

We consider the typical supervised learning task of predicting an output $y \in \mathcal{Y}$ from an input $x \in \mathcal{X}$, where the pairs $(x, y)$ are drawn from some unknown joint distribution, $\mathcal{D}$. The learning problem consists of inferring a function $h_S : \mathcal{X} \to \mathcal{Y}$ from a finite training dataset $S$ of $m$ i.i.d. samples from $\mathcal{D}$. The quality of a predictor $h$ can quantified by the expected error,

$$\mathcal{E}(h) = \mathbb{E}_{(x,y) \sim \mathcal{D}} \, \ell(h(x), y) \tag{1}$$

for some loss function $\ell : \mathcal{Y} \times \mathcal{Y} \to \mathbb{R}$.

In this paper, predictors $h_\theta$ are parametrized by the weights $\theta \in \mathbb{R}^N$ of deep neural networks. We consider a *frequentist risk* analysis of the learning algorithm, that is, the average performance over possible training sets (denoted by the random variable $S$) of size $m$. This is the same quantity Geman et al. (1992) consider. While $S$ is the only random quantity focused on in traditional bias-variance decomposition, we also focus on randomness coming from optimization. We denote the random variable for optimization randomness (e.g. initialization) by $I$.[2]

Formally, given a fixed training set $S$ and fixed optimization randomness $I$, the learning algorithm $\mathcal{A}$ produces $\theta = \mathcal{A}(S, I)$. Randomness in initialization translates to randomness in $\mathcal{A}(S, \cdot)$ because of non-convexity of the loss surface. Given a fixed training set, we encode the randomness due to $I$ in a conditional distribution $p(\theta|S)$; marginalizing over the training set $S$ of size $m$ gives a marginal distribution $p(\theta) = \mathbb{E}_S p(\theta|S)$ on the weights learned by $\mathcal{A}$ from $m$ samples. In this context, the frequentist risk for the learning algorithm using training sets of size $m$ becomes:

$$\mathcal{R}_m = \mathbb{E}_{\theta \sim p} \mathcal{E}(h_\theta) = \mathbb{E}_S \mathbb{E}_{\theta \sim p(\cdot|S)} \mathcal{E}(h_\theta) = \mathbb{E}_S \mathbb{E}_I \mathcal{E}(h_\theta) \tag{2}$$

### 2.2 BIAS-VARIANCE DECOMPOSITION

We briefly recall the standard bias-variance decomposition of the frequentist risk in the case of squared-losses. We work in the context of classification, where each class $k \in \{1 \cdots K\}$ is represented by a one-hot vector in $\mathbb{R}^K$. The predictor outputs a score or probability vector in $\mathbb{R}^k$. In this context, the risk in Eqn. 2 decomposes into three sources of error (Geman et al., 1992):

$$\mathcal{R}_m = \mathcal{E}_{\text{noise}} + \mathcal{E}_{\text{bias}} + \mathcal{E}_{\text{variance}} \tag{3}$$

The first term is an intrinsic error term independent of the predictor; the second is a bias term

$$\mathcal{E}_{\text{noise}} = \mathbb{E}_{(x,y)} \left[ \|y - \bar{y}(x)\|^2 \right], \qquad \mathcal{E}_{\text{bias}} = \mathbb{E}_x \left[ \|\mathbb{E}_\theta[h_\theta(x)] - \bar{y}(x)\|^2 \right], \tag{4}$$

where $\bar{y}(x)$ denotes the expectation $\mathbb{E}[y|x]$ of $y$ given $x$. The third term is the expected variance of the output predictions:

$$\mathcal{E}_{\text{variance}} = \mathbb{E}_x \text{Var}(h_\theta(x)), \qquad \text{Var}(h_\theta(x)) = \mathbb{E}_\theta \left[ \|(h_\theta(x) - \mathbb{E}_\theta[h_\theta(x)]\|^2 \right]$$

where the expectation over $\theta$ can be done as in Eqn. 2. Finally, in the set-up of Section 2.1, the sources of variance are the choice of training set $S$ and the choice of initialization $I$ (encoded into the conditional $p(\cdot|S)$). By the law of total variance, we then have the further decomposition:

$$\text{Var}(h_\theta(x)) = \mathbb{E}_S \left[ \text{Var}_I (h_\theta(x)|S) \right] + \text{Var}_S \left( \mathbb{E}_I [h_\theta(x)|S] \right) \tag{5}$$

We call the first term *variance due to initialization* and the second term *variance due to sampling* throughout the paper. Note that risks computed with classification losses (e.g cross-entropy or 0-1 loss) do not have such a clean bias-variance decomposition (Domingos, 2000; James, 2003). However, it is natural to expect that bias and variance are useful indicators of the performance of the models. In fact, the classification risk can be bounded as 4 times the regression risk (Appendix D.2).

---

[2]We do not study randomness from stochastic mini-batching because we found the phenomenon of decreasing variance with width persists when using *batch* gradient descent (Section 3.3, Appendix B.3).

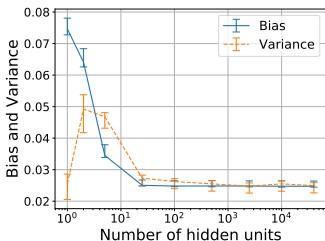 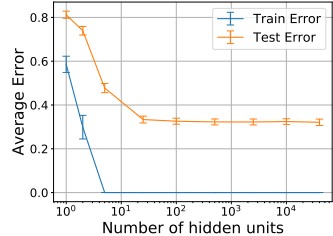 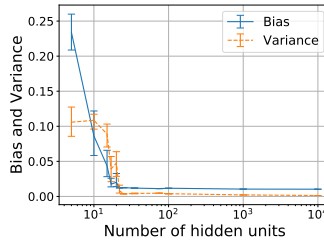

(a) Variance decreases with width, even in the small MNIST setting.

(b) Test error trend is same as bias-variance trend (small MNIST).

(c) Similar bias-variance trends on sinusoid regression task.

## 3 VARIANCE AND WIDTH

In this section, we study how variance of single hidden layer networks varies with width, like a modern analog of Geman et al. (1992). We study fully connected single hidden layer networks up to the largest size that fits in memory, in order to search for an eventual increase in variance. To make our study as general as possible, we consider networks without any regularization bells and whistles such as weight decay, dropout, or data augmentation, which Zhang et al. (2017) found to not be necessary for good generalization. As is commonly done in practice, these networks are trained with optimizers (e.g. SGD) whose step sizes are tuned using a validation set.[3]

### 3.1 COMMON EXPERIMENTAL DETAILS

Experiments are run on different datasets: full MNIST, small MNIST, and a sinusoid regression task. Averages over data samples are performed by taking the training set $S$ and creating 50 bootstrap (Efron, 1979) replicate training sets $S'$ by sampling with replacement from $S$. We train 50 different neural networks for each hidden layer size using these different training sets. Then, we estimate $\mathcal{E}_{\text{bias}}$ and $\mathcal{E}_{\text{variance}}$ as in Section 2.2, where the population expectation $\mathbb{E}_x$ is estimated with an average over the test set inputs.[4] To estimate the two terms from the law of total variance (Equation 5), we use 10 random seeds for the outer expectation and 10 for the inner expectation, resulting in a total of 100 seeds. Furthermore, we compute 99% confidence intervals for our bias and variance estimates using the bootstrap (Efron, 1979).

The networks are trained using SGD with momentum and generally run for long after 100% training set accuracy is reached (e.g. 500 epochs for full data MNIST and 10000 epochs for small data MNIST). The step size hyperparameter is fixed to 0.1 for the full data experiment and is chosen via a validation set for the small data experiment. The momentum hyperparameter is always set to 0.9.

### 3.2 DECREASING VARIANCE IN FULL DATA SETTING

We find a clear decreasing trend in variance with width of the network in the full data setting (Figure 1). The trend is the same with or without early stopping, so early stopping is not necessary to see decreasing variance, similar to how it was not necessary to see better test set performance with width in Neyshabur et al. (2014).

### 3.3 TESTING THE LIMITS: DECREASING VARIANCE IN THE SMALL DATA SETTING

Decreasing the size of the dataset can only increase variance. To study the robustness of the above observation, we decrease the size of the training set to just 100 examples. In this small data setting, somewhat surprisingly, we still observe the same trend of decreasing variance with width (Figure 3a). The test error behaves similarly (Figure 3b). The step size is tuned using a validation set (Appendix B.1). The training for tuning is stopped after 1000 epochs, whereas the training for the final models is stopped after 10000 epochs.

---

[3]Note that tuning the step size controls validation error for a specific network size. The question we study in our empirical analysis is how variance at these low validation error points varies with size of the network.

[4]Because we don't have access to $\bar{y}$, we use the labels $y$ to estimate $\mathcal{E}_{\text{bias}}$. This is equivalent to assuming noiseless labels and is standard procedure for estimating bias (Kohavi and Wolpert, 1996; Domingos, 2000).

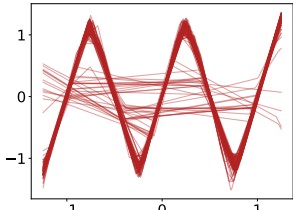 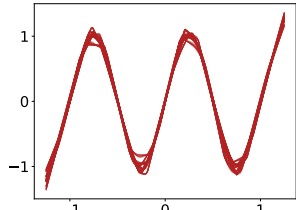 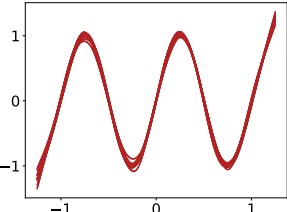

Figure 4: Visualization of the 100 different learned functions of single hidden layer neural networks of widths 15, 1000, and 10000 (from left to right) on the task of learning a sinusoid. The learned functions are increasingly similar with width, not increasingly different. More in Appendix B.4.

The corresponding experiment where step size is the same 0.01 for all network sizes is in Appendix B.2. With the same step size for all networks, we do not see decreasing variance. Note that we are not claiming that variance decreases with width regardless of step size. Rather, we are claiming variance decreases with width when the step size is tuned using a validation set, as is done in practice. By tuning the step size, we are making the experimental design choice of keeping *optimality of step size* constant across networks (more discussion on this in Appendix B.2).

This sensitivity to step size in the small data setting is evidence that we are testing the limits of our hypothesis. A larger amount of data makes the networks more robust to the choice of step size (Figure 1). However, it is likely the case that if we were able to compute with much larger networks, we would eventually observe increasing variance in the full data setting as well. By looking at the small data setting, we are able to test our hypothesis when the ratio of size of network to dataset size is quite large, and we still find this decreasing trend in variance (Figure 3a).

To see how dependent this phenomenon is on SGD, we also run these experiments using batch gradient descent and PyTorch's version of LBFGS. Interestingly, we find a decreasing variance trend with those optimizers as well. These experiments are included in Appendix B.3. This means that this decreasing variance phenomenon is not explained by the concept that "SGD implicitly regularizes."

### 3.4 DECOUPLING VARIANCE DUE TO SAMPLING FROM VARIANCE DUE TO INITIALIZATION

In order to better understand this variance phenomenon in neural networks, we separate the variance due to sampling from the variance due to initialization, according to the law of total variance (Equation 5). Contrary to what traditional bias-variance tradeoff intuition would suggest, we find variance due to sampling is roughly independent of width (Figure 2). Furthermore, we find that variance due to initialization decreases with width, causing the joint variance to decrease with width (Figure 2).

A body of recent work has provided evidence that over-parameterization (in width) helps gradient descent optimize to global minima in neural networks (Du et al., 2019; Du and Lee, 2018; Soltanolkotabi et al., 2017; Livni et al., 2014; Zhang et al., 2018). Always reaching a global minimum implies low variance due to initialization on the *training set*. Our observation of decreasing variance on the *test set* shows that the over-parameterization (in width) effect on optimization seems to extend to generalization, on the data sets we consider.

### 3.5 VISUALIZATION WITH REGRESSION ON SINUSOID

We trained different width neural networks on a noisy sinusoidal distribution with 80 independent training examples. This sinusoid regression setting also exhibits the familiar bias-variance trends (Figure 3c) and trends of the two components of the variance (Figure 5c).

Because this setting is low-dimensional, we can visualize the learned functions. The classic caricature of high capacity models is that they fit the training data in a very erratic way (example in Figure 11 of Appendix B.4). We find that wider networks learn sinusoidal functions that are much more similar than the functions learned by their narrower counterparts (Figure 4). We have analogous plots for all of the other widths and ones that visualize the variance similar to how it is commonly visualized for Gaussian processes in Appendix B.4.

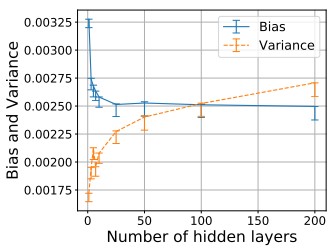 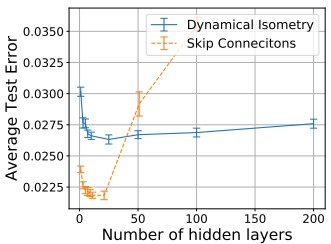 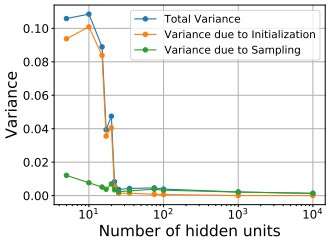

(a) Bias and variance trends with depth, using dynamical isometry

(b) Test error trends, using dynamical isometry vs. skip connections

(c) Similar trends with width of total variance terms on sinusoid regression task

# 4 VARIANCE AND DEPTH

In this section, we study the effect of depth on bias and variance by fixing width and varying depth. Historically, there have been pathological problems that cause deeper networks to experience higher test error than their shallower counterparts (Glorot and Bengio, 2010; He et al., 2016; Balduzzi et al., 2017). This indicates that there are some important confounding factors to control for when varying depth. The best control that we found is to use an initialization that achieves *dynamical isometry*, the condition that all of the singular values of the input-output Jacobian are 1 at initialization (Saxe et al., 2014; Pennington et al., 2017), as it allows networks to achieve test error that is nearly independent of depth (Figure 5b). See Appendix C.1 for more discussion on this.

## 4.1 TOTAL VARIANCE EXPERIMENTS

We train fully connected networks up to 200 layers deep and observe slowly increasing variance with depth (Figure 5a). The experimental protocol is similar to what it was in Section 3, with a few differences: All networks have width 100 and achieve 0 training error. We train them to the same loss value of 5e-5 to control for differences in training loss. This value was chosen carefully by observing when training error had been 0 for a long time. The 3 kinds of different networks we train are vanilla fully connected, fully connected with skip connections, and fully connected with dynamical isometry initialization. We only show the experiment with dynamical isometry in the main paper, but the other two are in Appendix C.2 and Appendix C.3.

We settle on fully connected networks without skip connections, initialized using the initialization Pennington et al. (2017) recommend to achieve dynamical isometry. This is the best experimental protocol of the three we tried because it appears to largely mitigate the pathological problems that cause deeper networks to have higher test error. We compare the test accuracy of skip connections to that of dynamical isometry in Figure 5b[5] to see that while the test accuracy of skip connections varies by over 1% from depth 25 to 100, the corresponding error bars for the dynamical isometry test errors overlap (although test error does increase by about 0.1% from depth 25 to 200). This near lack of dependence of test error on depth is why we view this experiment as having controlled for confounding factors sufficiently well. Additionally, this is the only protocol of the three we tested where bias monotonically decreases with depth (Figure 5a, Appendix C).

Just as new advancements, such as skip connections and dynamical isometry, have greatly helped with test set performance, there could still be future advancements that change these results. For example, it seems plausible that we will eventually have model families whose test error decreases (with depth) until it plateaus and, similarly, variance that increases and plateaus.

---

[5]Note that the best dynamical isometry network achieves test set accuracy of about 0.4% worse than the best skip connection network due to the fact that dynamical isometry is not possible with ReLU activations, so Tanh is used (Pennington et al., 2017).

### 4.2 DECOUPLING VARIANCE DUE TO SAMPLING FROM VARIANCE DUE TO INITIALIZATION

To get a more fine-grain look at the effect of depth on variance, we estimate the terms of the law of total variance in Figure 2, just as we did for width. Surprisingly, variance due to sampling is roughly constant again. Variance due to initialization increases with depth.

We view the increase in variance due to initialization that we observe as consistent with the conventional wisdom that Arora et al. (2018) summarizes: "Conventional wisdom in deep learning states that increasing depth improves expressiveness but complicates optimization." While Arora et al. (2018) focus on speed of training, the variance we measure in Figure 2 is about the diversity of different minima. Increasing depth seems to lead to different initial starting points optimizing to increasingly different functions, as evaluated on the *test set*. Li et al. (2017, Figure 7) provide visualizations that suggest that increasing depth leads to increasingly "chaotic" loss landscapes, which would indicate increasing variance on the training set.

## 5 DISCUSSION AND THEORETICAL INSIGHTS FOR INCREASING WIDTH

Our empirical results demonstrate that in the practical setting, variance due to initialization decreases with network width while variance due to sampling remains constant. In Section 5.1, we review classical results from linear models and remark that these trends can be seen in over-parameterized linear models. In Section 5.2 we take inspiration from linear models to provide analogous arguments for this phenomenon in increasingly wide neural networks, under strong assumptions. In Section 5.3, we note the mismatch between width and depth (the trend of variance due to initialization with width is opposite the corresponding trend with depth), and we discuss why the assumptions in Section 5.2 might be increasingly inaccurate with deeper and deeper networks.

### 5.1 INSIGHTS FROM LINEAR MODELS

The goal here is to gain insights from simple linear models. We discuss the standard setting which assumes a noisy linear mapping $y = \theta^T x + \epsilon$ between input feature vectors $x \in \mathbb{R}^N$ and real outputs, where $\mathbb{E}(\epsilon) = 0$ and $\text{Var}(\epsilon) = \sigma_\epsilon^2$. Note that $x$ is not necessarily raw data, but can be thought of as the embedding of the raw data in $\mathbb{R}^N$, using feature functions; this allows for the "over-parameterized" setting in linear models where $N > m$, regardless of the dimensionality of the raw data. We consider linear fits $\hat{y} = \hat{\theta}^T x$ obtained using mean-square error gradient-descent with random initialization.

We revisit the standard variance analysis for linear regression (Hastie et al., 2009, Section 7.3), where one can give the explicit form of the gradient descent solution. For a training set $S$ of size $m$, let $X_S$ denote the $m \times N$ data matrix whose $i^{\text{th}}$ row is the training point $x_i^T$. We also introduce the input correlation matrices:

$$\Sigma_S = X_S^T X_S, \qquad \Sigma = \mathbb{E}_x[xx^T] \tag{6}$$

The case where $N \leq m$ is standard: if $X_S$ has maximal rank, $\Sigma_S$ is invertible; the solution is independent of the initialization and given by

$$\hat{\theta}_S = \theta + \Sigma_S^{-1} X_S^T \epsilon \tag{7}$$

In the "fixed design" scenario, where we consider fixed training points $x_i$, the expected prediction variance with respect to noise is then

$$\mathbb{E}_x \text{Var}_\epsilon(\hat{y}) = \sigma_\epsilon^2 \text{Tr}[\Sigma \Sigma_S^{-1}] \tag{8}$$

In this case, the variance grows with the number of parameters. For example, by replacing $\Sigma$ with its unbiased estimator $m^{-1}\Sigma_S$, we recover the standard value $(N/m)\sigma_\epsilon^2$ (Hastie et al., 2009).

The "over-parametrized" case where $N > m$ is more interesting: even if $X_S$ has maximal rank, $\Sigma_S$ is not invertible. The kernel of $\Sigma_S$ is the subspace $U_S^\perp$ orthogonal to the span $U_S$ of the training points $x_i$. Gradient descent updates belong to $U_S$, independent of $U_S^\perp$. Initialized at $\theta_0$, it gives the solution

$$\hat{\theta}_S = P_{S^\perp}(\theta_0) + P_S(\theta) + \Sigma_S^+ X_S^T \epsilon \tag{9}$$

where $P_S$ and $P_{S^\perp}$ are the projections onto $U_S$ and $U_S^\perp$, and superscript $+$ denotes the pseudo-inverse. The first term, orthogonal to the data, does not get updated during training and only depends on the initialization. The two others form the minimum norm solution, which lies in $U_S$.

The form of the solution (Equation 9) has several consequences:

(a) Initialization contributes to the variance. Thus, for the input $x$ and using a standard initialization[6] $\theta_0 \sim \mathcal{N}(0, \frac{1}{N}I)$, we obtain

$$\mathrm{Var}_{\theta_0}(\hat{y}_S) = \frac{1}{N}\|P_{S^\perp}(x)\|^2 \tag{10}$$

which is non zero whenever $x$ has components orthogonal to the training data. Note, however, that the variance due to initialization actually *decreases* with the number of parameters.

(b) The expected variance due to noise is

$$\mathbb{E}_x \mathrm{Var}(\hat{y}) = \sigma_\epsilon^2 \mathrm{Tr}[\Sigma\Sigma_S^+] \tag{11}$$

In this case, the variance *scales as the dimension of the data*, as opposed to the number of parameters. Thus, replacing $\Sigma$ by its unbiased estimator $m^{-1}\Sigma_S$, we find the value $(r/m)\sigma_\epsilon^2$ where $r = \mathrm{rank}(\Sigma_S) = \dim U_S$.

We argue in the next section that, under specific assumptions that we discuss, these insights may be relevant for the non-linear case.

## 5.2 A MORE GENERAL RESULT

We will illustrate our arguments in the following simplified setting.

**Setting.** Let $N$ be the dimension of the parameter space. The prediction for a fixed example $x$, given by a trained network parameterized by $\theta$ depends on:

(i) a subspace of the parameter space, $\mathcal{M} \in \mathbb{R}^N$ with relatively small dimension, $d(N)$, which depends only on the learning task.

(ii) parameter components corresponding to directions orthogonal to $\mathcal{M}$. The orthogonal $\mathcal{M}^\perp$ of $\mathcal{M}$ has dimension, $N - d(N)$, and is essentially irrelevant to the learning task.

We can write the parameter vector as a sum of these two components $\theta = \theta_\mathcal{M} + \theta_{\mathcal{M}^\perp}$. We will further make the following assumptions.

(a) The optimization of the loss function is invariant with respect to $\theta_{\mathcal{M}\perp}$.

(b) Regardless of initialization, the optimization method consistently yields a solution with the same $\theta_\mathcal{M}$ component, (i.e. the same vector when projected onto $\mathcal{M}$).

These are strong assumptions, but there is some support for them in the literature. Li et al. (2018) empirically showed the existence of a critical number $d(N) = d$ of relevant parameters for a given learning task, independent of the size of the model. Sagun et al. (2017) showed that the spectrum of the Hessian for over-parametrized networks splits into $(i)$ a bulk centered near zero and $(ii)$ a small number of large eigenvalues, which suggests[7] that learning occurs mainly in a small number of directions. The existence of a subspace $\mathcal{M}_\perp$ in which no learning occurs was also conjectured by Advani and Saxe (2017) and shown to hold in deep linear networks under a simplifying assumption that decouples the dynamics of the weights in different layers.

### 5.2.1 VARIANCE DUE TO INITIALIZATION

Given the above assumptions, the following result shows that the variance from initialization vanishes as we increase $N$. The full proof, which builds on concentration results for Gaussians (based on Levy's lemma (Ledoux, 2001)), is given in Appendix D.

**Theorem 1** (Decay of variance due to initialization)**.** *Consider the setting of Section 5.2 Let $\theta$ denote the parameters at the end of the learning process. Then, for a fixed data set and parameters initialized as $\theta_0 \sim \mathcal{N}(0, \frac{1}{N}I)$, the variance of the prediction satisfies the inequality,*

$$Var_{\theta_0}(h_\theta(x)) \leq C\frac{2L^2}{N} \tag{12}$$

*where $L$ is the Lipschitz constant of the prediction with respect to $\theta$, and for some universal constant $C > O$.*

---

[6]It is such that the initial parameter norm $\|\theta_0\|$ has unit variance.

[7]Provided the corresponding eigenspace decomposition is preserved throughout training.

This result guarantees that the variance decreases to zero as $N$ increases, provided the Lipschitz constant $L$ grows more slowly than the square root of dimension, $L = o(\sqrt{N})$.

### 5.2.2 VARIANCE DUE TO SAMPLING

Under the above assumptions, the parameters at the end of learning take the form $\theta = \theta_{\mathcal{M}}^* + \theta_{0\mathcal{M}^\perp}$. For fixed initialization, the only source of variance of the prediction is the randomness of $\theta_{\mathcal{M}}^*$ on the learning manifold. The variance depends on the parameter dimensionality only through $\dim \mathcal{M} = d(N)$, and hence remains constant if $d(N)$ does (Li et al., 2018).

### 5.3 DISCUSSION ON ASSUMPTIONS IN INCREASINGLY DEEP NETWORKS

The mismatch between the outcome of our theoretical analysis and the observed trend of variance due to initialization with depth suggests that our assumptions are increasingly inaccurate with depth. For some intuition about why this may be the case, consider the dependence of gradients with respect to subsets of hidden units, as these gradients are related to assumption (a): the invariance of the optimization process to $\theta_{\mathcal{M}^\perp}$. Gradients of hidden units in the same layer (related to width) do not directly depend on each other; rather, only optimization induces dependencies between them via the loss function. In sharp contrast, hidden units in different layers (related to depth) are functions of their preceding layers, and similarly, the gradients with respect to earlier layers are *functionally dependent* on the gradients with respect to later layers. This hints at more complex optimization interactions between parameters when increasing depth.

## 6 CONCLUSION AND FUTURE WORK

By revisiting the bias-variance decomposition and using a finer-grain method to empirically study variance, we find interesting phenomena. First, the bias-variance tradeoff is misleading for network width (one way to increase size) as the measure of model complexity. Second, variance due to sampling does not appear to be dependent on width or depth. Third, variance due to initialization is roughly consistent with the optimization literature, as we observe the test set analog of the current conventional wisdom for both width and depth. Finally, by taking inspiration from linear models, we perform a theoretical analysis of the variance that is consistent with our empirical observations for increasing width.

We view future work that uses the bias-variance lens as promising. For example, a probabilistic notion of effective capacity of a model is natural when studying generalization through this lens (Appendix A). We did not study how bias and variance change over the course of training; that would make an interesting direction for future work. Additionally, it may be fruitful to apply the bias-variance lens to other network architectures, such as convolutional networks and recurrent networks. We argue it is worth running variance vs. depth experiments using future best practices to train deep models, as the results could be different. More theoretical work is also needed to achieve a full understanding of the behaviour of variance in deep models. Variance is analytically different from generalization error in that the definition of variance does not involve the labels at all. We view the bias-variance lens as a useful tool for studying generalization in deep learning and hope to encourage more work in this direction.

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

# Appendices

## APPENDIX A    PROBABILISTIC NOTION OF EFFECTIVE CAPACITY

The problem with classical complexity measures is that they do not take into account optimization and have no notion of what will actually be learned. Arpit et al. (2017, Section 1) define a notion of an *effective* hypothesis class to take into account what functions are possible to be learned by the learning algorithm.

However, this still has the problem of not taking into account what hypotheses are *likely* to be learned. To take into account the probabilistic nature of learning, we define the *ϵ-hypothesis class* for a data distribution $\mathcal{D}$ and learning algorithm $\mathcal{A}$, that contains the hypotheses which are at least $\epsilon$-likely for some $\epsilon > 0$:

$$\mathcal{H}_\mathcal{D}(\mathcal{A}) = \{h : p(h(\mathcal{A}, S)) \geq \epsilon\}, \tag{13}$$

where $S$ is a training set drawn from $\mathcal{D}^m$, $h(\mathcal{A}, S)$ is a random variable drawn from the distribution over learned functions induced by $\mathcal{D}$ and the randomness in $\mathcal{A}$; $p$ is the corresponding density. Thinking about a model's $\epsilon$-hypothesis class can lead to drastically different intuitions for the complexity of a model and its variance (Figure 6). This is at the core of the intuition for why the traditional view of bias-variance as a tradeoff does not hold in all cases.

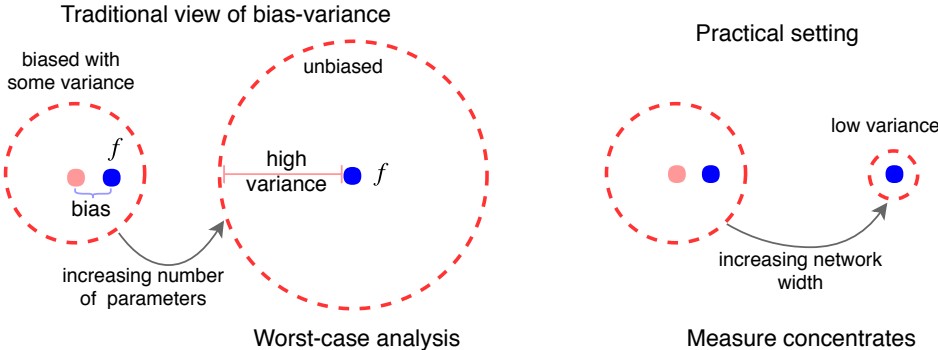

Figure 6: The dotted red circle depicts a cartoon version of the $\epsilon$-hypothesis class of the learner. The left side reflects common intuition, as informed by the bias-variance tradeoff and worst-case analysis from statistical learning theory. The right side reflects our view that variance can decrease with network width.

## APPENDIX B    WIDTH AND VARIANCE: ADDITIONAL EMPIRICAL RESULTS AND DISCUSSION

### B.1    TUNED LEARNING RATES FOR SGD

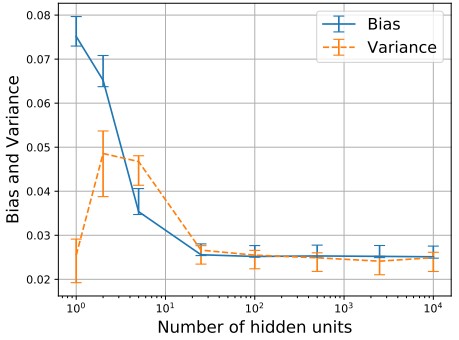

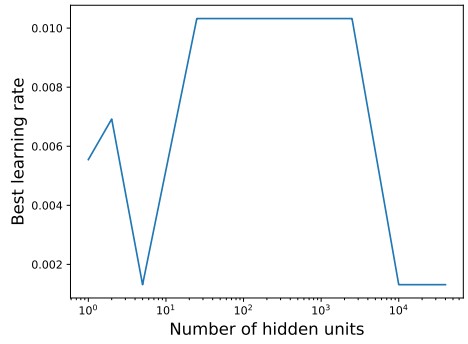

(a) Variance decreases with width, even in the small data setting (SGD). This figure is in the main paper, but we include it here to compare with the corresponding step sizes used.

(b) Corresponding optimal learning rates found, by random search, and used.

### B.2    FIXED LEARNING RATE RESULTS FOR SMALL DATA MNIST

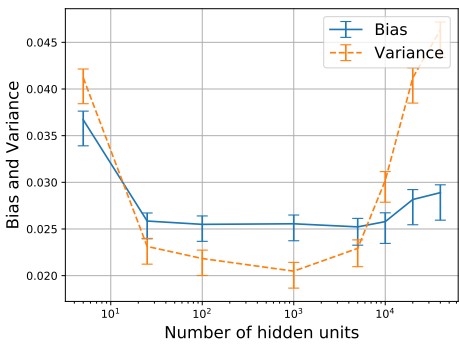

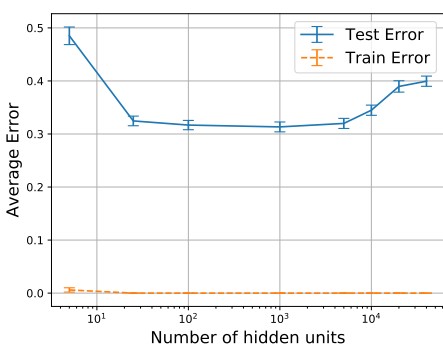

Figure 8: Variance on small data with a fixed learning rate of 0.01 for all networks.

Note that the U curve shown in Figure 8 when we do not tune the step size is explained by the fact that the constant step chosen is a "good" step size for some networks and "bad" for others. Results from Keskar et al. (2017) and Smith et al. (2018) show that a step size that corresponds well to the noise structure in SGD is important for achieving good test set accuracy. Because our networks are different sizes, their stochastic optimization process will have a different landscape and noise structure. By tuning the step size, we are making the experimental design choice to keep *optimality of step size* constant across networks, rather than keeping step size constant across networks. To us, choosing this control makes much more sense than choosing to control for step size.

### B.3 Other optimizers for width experiment on small data MNIST

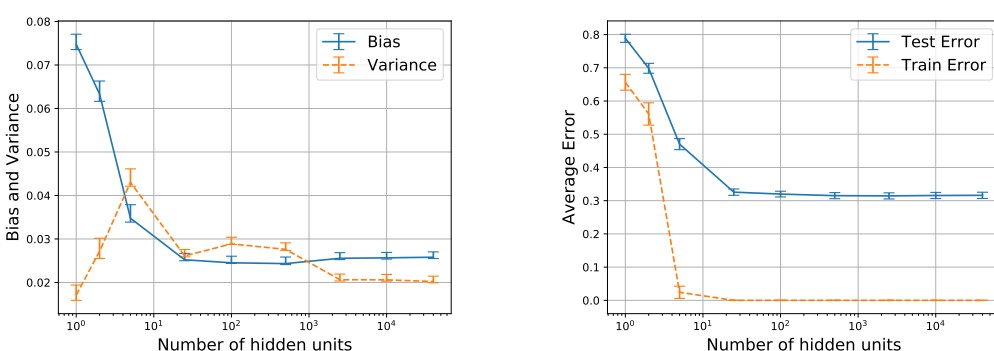

Figure 9: Variance decreases with width in the small data setting, even when using batch gradient descent.

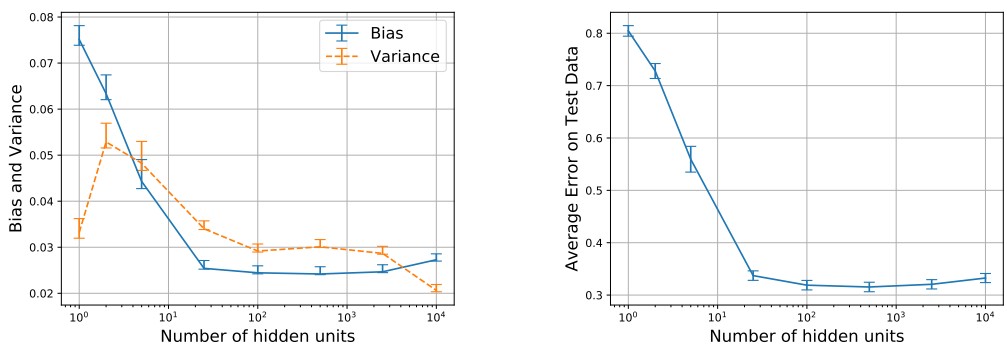

Figure 10: Variance decreases with width in the small data setting, even when using a strong optimizer, such as PyTorch's LBFGS, as the optimizer.

### B.4 Sinusoid regression experiments

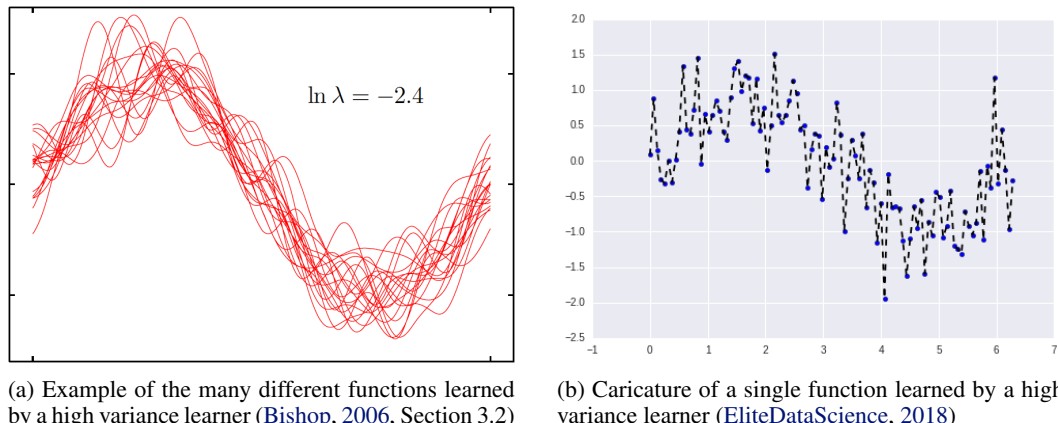

(a) Example of the many different functions learned by a high variance learner (Bishop, 2006, Section 3.2)

(b) Caricature of a single function learned by a high variance learner (EliteDataScience, 2018)

Figure 11: Caricature examples of high variance learners on sinusoid task. Below, we find that this does not happen with increasingly wide neural networks (Figure 13 and Figure 14).

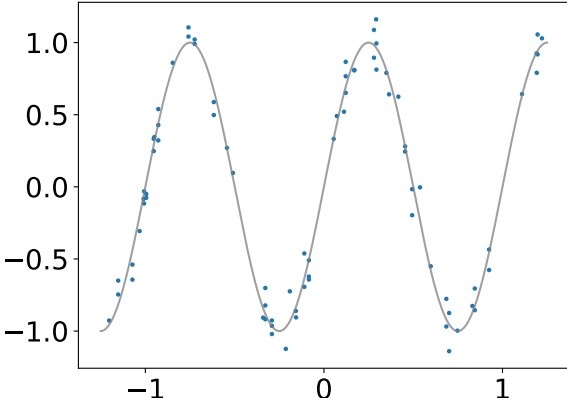

Figure 12: Target function of the noisy sinusoid regression task (in gray) and an example of a training set (80 data points) sampled from the noisy distribution.

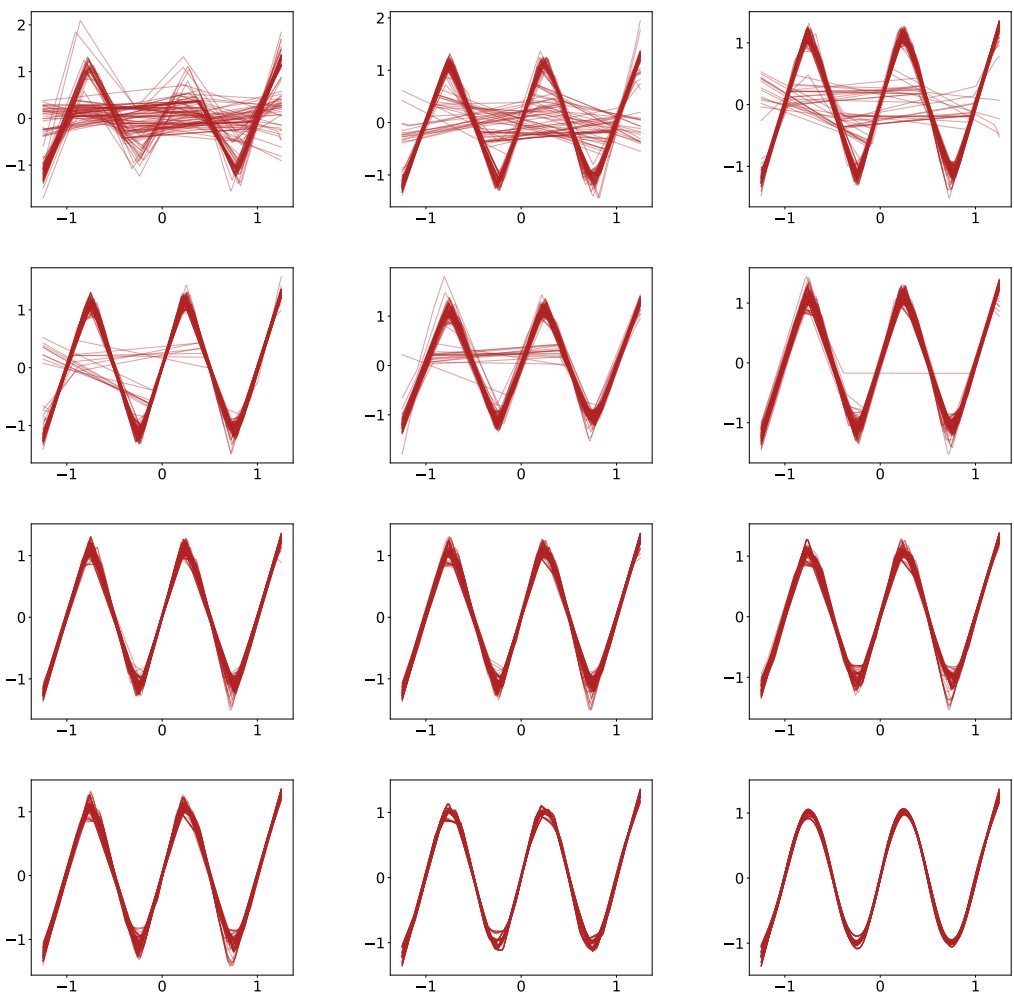

Figure 13: Visualization of 100 different functions learned by the different width neural networks. Darker color indicates higher density of different functions. Widths in increasing order from left to right and top to bottom: 5, 10, 15, 17, 20, 22, 25, 35, 75, 100, 1000, 10000. We do *not* observe the caricature from Figure 11 as width is increased.

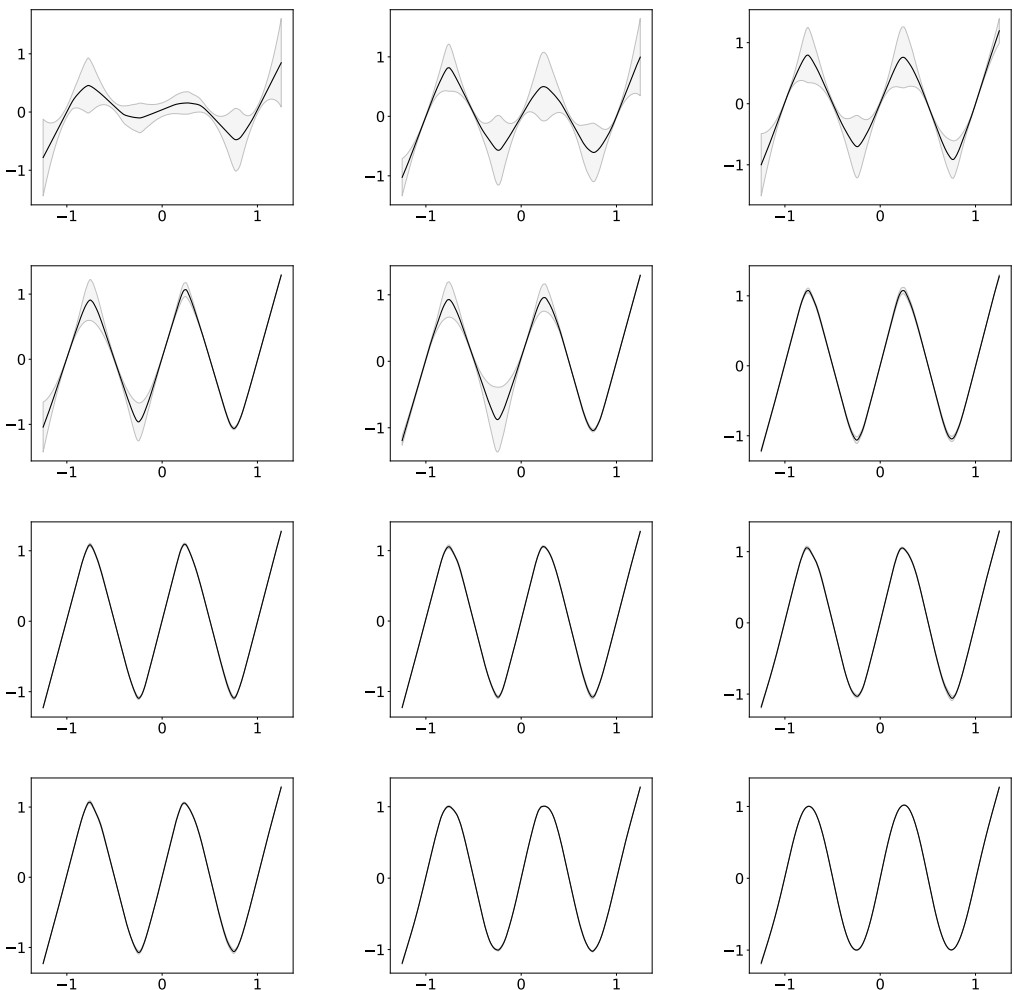

Figure 14: Visualization of the mean prediction and variance of the different width neural networks. Widths in increasing order from left to right and top to bottom: 5, 10, 15, 17, 20, 22, 25, 35, 75, 100, 1000, 10000.

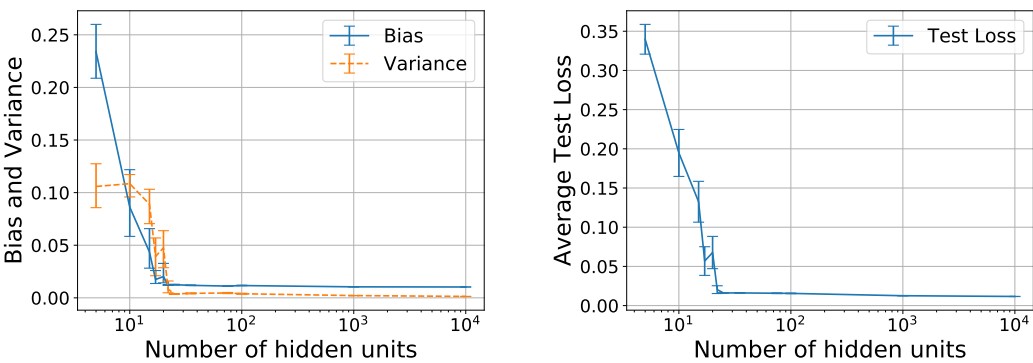

Figure 15: We observe the same trends of bias and total variance in the sinusoid regression setting. The figure on the left is in the main paper, while the figure on the right is support.

## APPENDIX C   DEPTH AND VARIANCE: ADDITIONAL EMPIRICAL RESULTS AND DISCUSSION

### C.1   DISCUSSION ON NEED FOR CAREFUL EXPERIMENTAL DESIGN

Depth is an important component of deep learning. We study its effect on bias and variance by fixing width and varying depth. However, there are pathological problems associated with training very deep networks such as vanishing/exploding gradient (Hochreiter, 1991; Bengio et al., 1994; Glorot and Bengio, 2010), signal not being able to propagate through through the network (Schoenholz et al., 2017), and gradients resembling white noise (Balduzzi et al., 2017). He et al. (2016) pointed out that very deep networks experience high test set error and argued it was due to high training set loss. However, while skip connections (He et al., 2016), better initialization (Glorot and Bengio, 2010), and batch normalization (Ioffe and Szegedy, 2015) have largely served to facilitate low training loss in very deep networks, the problem of high *test set* error still remains.

The current best practices for achieving low test error in very deep networks arose out of trying to solve the above problems in training. An initial step was to ensure the mean squared singular value of the input-output Jacobian, at initialization, is close to 1 (Glorot and Bengio, 2010). More recently, there has been work on a stronger condition known as *dynamical isometry*, where *all* singular values remain close to 1 (Saxe et al., 2014; Pennington et al., 2017). Pennington et al. (2017) also empirically found that dynamical isometry helped achieve low test set error. Furthermore, Xiao et al. (2018, Figure 1) found evidence that test set performance did not degrade with depth when they lifted dynamical isometry to CNNs. This why we settled on dynamical isometry as the best known practice to control for as many confounding factors as possible.

We first ran experiments with vanilla full connected networks (Figure 16). These have clear training issues where networks of depth more than 20 take very long to train to the target training loss of 5e-5. The bias curve is not even monotonically decreasing. Clearly, there are important confounding factors not controlled for in this simple setting. Still, note that variance increases roughly linearly with depth.

We then study fully connected networks with skip connections between every 2 layers (Figure 17). While this allows us to train deeper networks than without skip connections, many of the same issues persist (e.g. bias still not monotonically decreasing). The bias, variance, and test error curves are all checkmark-shaped.

### C.2   VANILLA FULLY CONNECTED DEPTH EXPERIMENTS

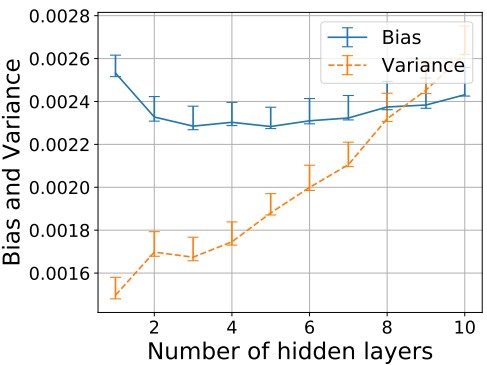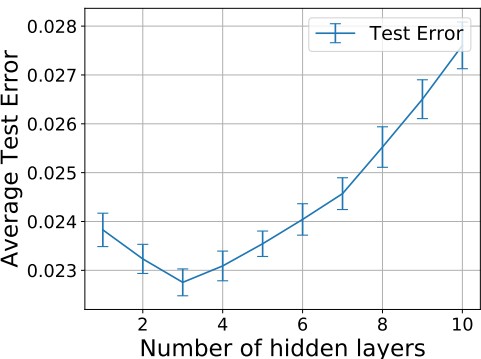

Figure 16: Test error quickly degrades in fairly shallow fully connected networks, and bias does not even monotonically decrease with depth. However, this is the first indication that variance might *increase* with depth. All networks have training error 0 and are trained to the same training loss of 5e-5.

## C.3 Skip connections depth experiments

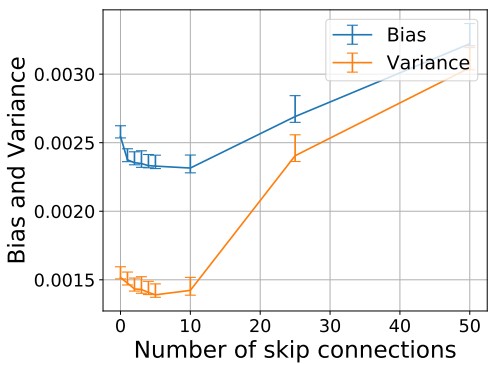
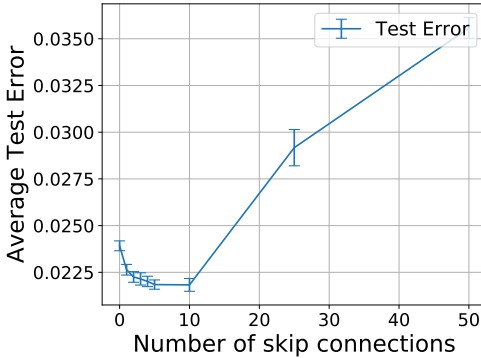

Figure 17: While the addition of skip connections (between every other layer) might push the bottom of the U curve in test error out to 10 skip connections (21 layers), which is further than 3 layers, which is what was seen without skip connections, test error still degrades noticeably in greater depths. Additionally, bias still does not even monotonically decrease with depth. While skip connections appear to have helped control for the factors we want to control, they were not completely satisfying. All networks have training error 0 and are trained to the same training loss of 5e-5.

## C.4 Dynamical isometry depth experiments

The figures in this section are included in the main paper, but they are included here for comparison to the above and for completeness.

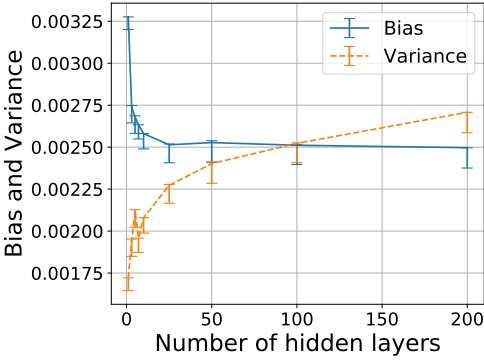
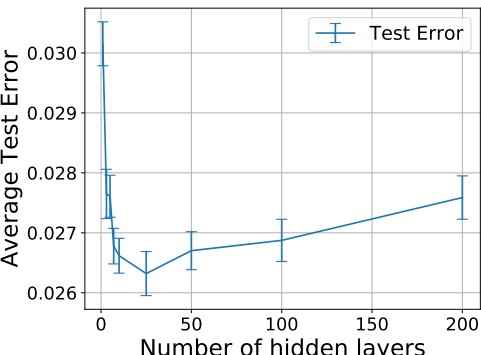

Figure 18: Additionally, dynamical isometry seems to cause bias to decrease monotonically with depth. While skip connections appear to have helped control for the factors we want to control, they were not completely satisfying. All networks have training error 0 and are trained to the same training loss of 5e-5.

## Appendix D    Some Proofs

### D.1    Proof of Theorem 1

First we state some known concentration results (Ledoux, 2001) that we will use in the proof.

**Lemma 1** (Levy). *Let $h : S_R^n \to \mathbb{R}$ be a function on the $n$-dimensional Euclidean sphere of radius $R$, with Lipschitz constant $L$; and $\theta \in S_R^n$ chosen uniformly at random for the normalized measure. Then*

$$\mathbb{P}(|h(\theta) - \mathbb{E}[h]| > \epsilon) \leq 2 \exp\left(-C \frac{n\epsilon^2}{L^2 R^2}\right) \tag{14}$$

*for some universal constant $C > 0$.*

Uniform measures on high dimensional spheres approximate Gaussian distributions (Ledoux, 2001). Using this, Levy's lemma yields an analogous concentration inequality for functions of Gaussian variables:

**Lemma 2** (Gaussian concentration). *Let $h : \mathbb{R}^n \to \mathbb{R}$ be a function on the Euclidean space $\mathbb{R}^n$, with Lipschitz constant $L$; and $\theta \sim \mathcal{N}(0, \sigma\mathbb{I}_n)$ sampled from an isotropic $n$-dimensional Gaussian. Then:*

$$\mathbb{P}(|h(\theta) - \mathbb{E}[h]| > \epsilon) \leq 2\exp\left(-C\frac{\epsilon^2}{L^2\sigma^2}\right) \tag{15}$$

*for some universal constant $C > 0$.*

Note that in the Gaussian case, the bound is dimension free.

In turn, concentration inequalities give variance bounds for functions of random variables.

**Corollary 1.** *Let $h$ be a function satisfying the conditions of Theorem 2, and $Var(h) = \mathbb{E}[(h - \mathbb{E}[h])^2]$. Then*

$$Var(h) \leq \frac{2L^2\sigma^2}{C} \tag{16}$$

*Proof.* Let $g = h - \mathbb{E}[h]$. Then $\text{Var}(h) = \text{Var}(g)$ and

$$\text{Var}(g) = \mathbb{E}[|g|^2] = 2\mathbb{E}\int_0^{|g|} t\,dt = 2\mathbb{E}\int_0^\infty t\mathbb{1}_{|g|>t}\,dt \tag{17}$$

Now swapping expectation and integral (by Fubini theorem), and by using the identity $\mathbb{E}\mathbb{1}_{|g|>t} = \mathbb{P}(|g| > t)$, we obtain

$$\begin{aligned}
\text{Var}(g) &= 2\int_0^\infty t\,\mathbb{P}_R(|g| > t)\,dt \\
&\leq 2\int_0^\infty 2t\exp\left(-C\frac{t^2}{L^2\sigma^2}\right)dt \\
&= 2\left[-\frac{L^2\sigma^2}{C}\exp\left(-C\frac{t^2}{L^2\sigma^2}\right)\right]_0^\infty = \frac{2L^2\sigma^2}{C}
\end{aligned}$$

$\square$

We are now ready to prove Theorem 1. We first recall our assumptions.

**Assumption 1.** *The optimization of the loss function is invariant with respect to $\theta_{\mathcal{M}\perp}$.*

**Assumption 2.** *Along $\mathcal{M}$, optimization yields solutions independently of the initialization $\theta_0$.*

We add the following assumptions.

**Assumption 3.** *The prediction $h_\theta(x)$ is $L$-Lipschitz with respect to $\theta_{\mathcal{M}\perp}$.*

**Assumption 4.** *The network parameters are initialized as*

$$\theta_0 \sim \mathcal{N}(0, \frac{1}{N} \cdot I_{N\times N}). \tag{18}$$

We first prove that the Gaussian concentration theorem translates into concentration of predictions in the setting of Section 5.2.1.

**Theorem 2** (Concentration of predictions). *Consider the setting of Section 5.2 and Assumptions 1 and 4. Let $\theta$ denote the parameters at the end of the learning process. Then, for a fixed data set, $S$ we get concentration of the prediction, under initialization randomness,*

$$\mathbb{P}(|h_\theta(x) - \mathbb{E}[h_\theta(x)]| > \epsilon) \leq 2\exp\left(-C\frac{N\epsilon^2}{L^2}\right) \tag{19}$$

*for some universal constant $C > 0$.*

*Proof.* In our setting, the parameters at the end of learning can be expressed as

$$\theta = \theta_{\mathcal{M}}^* + \theta_{\mathcal{M}^\perp} \tag{20}$$

where $\theta_{\mathcal{M}}^*$ is independent of the initialization $\theta_0$. To simplify notation, we will assume that, at least locally around $\theta_{\mathcal{M}}^*$, $\mathcal{M}$ is spanned by the first $d(N)$ standard basis vectors, and $\mathcal{M}^\perp$ by the remaining $N - d(N)$. This will allow us, from now on, to use the same variable names for $\theta_{\mathcal{M}}$ and $\theta_{\mathcal{M}^\perp}$ to denote their lower-dimensional representations of dimension $d(N)$ and $N - d(N)$ respectively. More generally, we can assume that there is a mapping from $\theta_{\mathcal{M}}$ and $\theta_{\mathcal{M}^\perp}$ to those lower-dimensional representations.

From Assumptions 1 and 4 we get

$$\theta_{\mathcal{M}^\perp} \sim \mathcal{N}\left(0, \frac{1}{N} I_{(N-d(N)) \times (N-d(N))}\right). \tag{21}$$

Let $g(\theta_{\mathcal{M}^\perp}) \triangleq h_{\theta_{\mathcal{M}}^* + \theta_{\mathcal{M}^\perp}}(x)$. By Assumption 3, $g(\cdot)$ is $L$-Lipschitz. Then, by the Gaussian concentration theorem we get,

$$\mathbb{P}(|g(\theta_{\mathcal{M}^\perp}) - \mathbb{E}[g(\theta_{\mathcal{M}^\perp})]| > \epsilon) \leq 2 \exp\left(-C \frac{N\epsilon^2}{L^2}\right). \tag{22}$$

$\square$

The result of Theorem 1 immediately follows from Theorem 2 and Corollary 1, with $\sigma^2 = 1/N$:

$$\text{Var}_{\theta_0}(h_\theta(x)) \leq C \frac{2L^2}{N} \tag{23}$$

Provided the Lipschitz constant $L$ of the prediction grows more slowly than the square of dimension, $L = o(\sqrt{N})$, we conclude that the variance vanishes to zero as $N$ grows.

## D.2   BOUND ON CLASSIFICATION ERROR IN TERMS OF REGRESSION ERROR

In this section we give a bound on classification risk $\mathcal{R}_{\text{classif}}$ in terms of the regression risk $\mathcal{R}_{\text{reg}}$.

**Notation.** Our classifier defines a map $h : \mathcal{X} \to \mathbb{R}^k$, which outputs probability vectors $h(x) \in \mathbb{R}^k$, with $\sum_{y=1}^k h(x)_y = 1$. The classification loss is defined by

$$L(h) = \text{Prob}_{x,y}\{h(x)_y < \max_{y'} h(x)_{y'}\}$$
$$= \mathbb{E}_{(x,y)} I(h(x)_y < \max_{y'} h(x)_{y'}) \tag{24}$$

where $I(a) = 1$ if predicate $a$ is true and 0 otherwise. Given trained predictors $h_S$ indexed by training dataset $S$, the classification and regression risks are given by,

$$\mathcal{R}_{\text{classif}} = \mathbb{E}_S L(h_S), \qquad \mathcal{R}_{\text{reg}} = \mathbb{E}_S \mathbb{E}_{(x,y)} ||h_S(x) - Y||_2^2 \tag{25}$$

where $Y$ denotes the one-hot vector representation of the class $y$.

**Proposition 1.** *The classification risk is bounded by four times the regression risk, $\mathcal{R}_{classif} \leq 4\mathcal{R}_{reg}$.*

*Proof.* First note that, if $h(x) \in \mathbb{R}^k$ is a probability vector, then

$$h(x)_y < \max_{y'} h(x)_{y'} \implies h(x)_y < \frac{1}{2}$$

By taking the expectation over $x, y$, we obtain the inequality $L(h) \leq \widetilde{L}(h)$ where

$$\widetilde{L}(h) = \text{Prob}_{x,y}\{h(x)_y < \frac{1}{2}\} \tag{26}$$

We then have,

$$
\begin{aligned}
\mathcal{R}_{\text{classif}} := \mathbb{E}_S L(h_S) &\leq \mathbb{E}_S \tilde{L}(h_S) \\
&= \text{Prob}_{S;\, x,y}\{h_S(x)_y < \frac{1}{2}\} \\
&= \text{Prob}_{S;\, x,y}\{|h_S(x)_y - Y_y| > \frac{1}{2}\} \\
&\leq \text{Prob}_{S;\, x,y}\{||h_S(x) - Y||_2 > \frac{1}{2}\} \\
&= \text{Prob}_{S;\, x,y}\{||h_S(x) - Y||_2^2 > \frac{1}{4}\} \leq 4\mathcal{R}_{\text{reg}}
\end{aligned}
$$

where the last inequality follows from Markov's inequality.

$\square$

## APPENDIX E    COMMON INTUITIONS FROM IMPACTFUL WORKS

"Neural Networks and the Bias/Variance Dilemma" from (Geman et al., 1992): "How big a network should we employ? A small network, with say one hidden unit, is likely to be biased, since the repertoire of available functions spanned by $f(x; w)$ over allowable weights will in this case be quite limited. If the true regression is poorly approximated within this class, there will necessarily be a substantial bias. On the other hand, if we overparameterize, via a large number of hidden units and associated weights, then the bias will be reduced (indeed, with enough weights and hidden units, the network will interpolate the data), but there is then the danger of a significant variance contribution to the mean-squared error. (This may actually be mitigated by incomplete convergence of the minimization algorithm, as we shall see in Section 3.5.5.)"

"An Overview of Statistical Learning Theory" from (Vapnik, 1999): "To avoid over fitting (to get a small confidence interval) one has to construct networks with small VC-dimension."

"Stability and Generalization" from Bousquet and Elisseeff (2002): "It has long been known that when trying to estimate an unknown function from data, one needs to find a tradeoff between bias and variance. Indeed, on one hand, it is natural to use the largest model in order to be able to approximate any function, while on the other hand, if the model is too large, then the estimation of the best function in the model will be harder given a restricted amount of data." Footnote: "We deliberately do not provide a precise definition of bias and variance and resort to common intuition about these notions."

Pattern Recognition and Machine Learning from Bishop (2006): "Our goal is to minimize the expected loss, which we have decomposed into the sum of a (squared) bias, a variance, and a constant noise term. As we shall see, there is a trade-off between bias and variance, with very flexible models having low bias and high variance, and relatively rigid models having high bias and low variance."

"Understanding the Bias-Variance Tradeoff" from Fortmann-Roe (2012): "At its root, dealing with bias and variance is really about dealing with over- and under-fitting. Bias is reduced and variance is increased in relation to model complexity. As more and more parameters are added to a model, the complexity of the model rises and variance becomes our primary concern while bias steadily falls. For example, as more polynomial terms are added to a linear regression, the greater the resulting model's complexity will be."

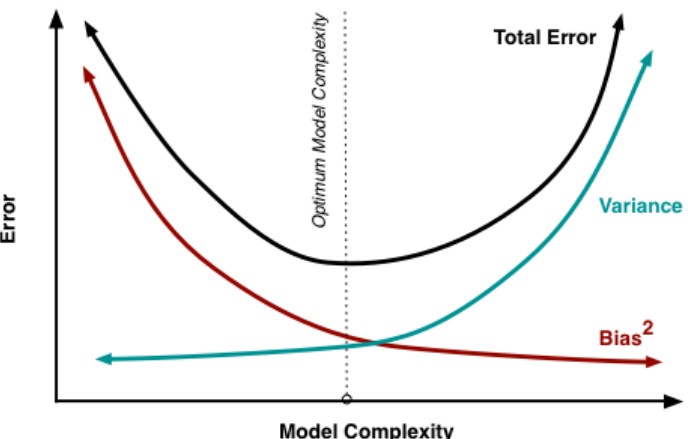

Figure 19: Illustration of common intuition for bias-variance tradeoff (Fortmann-Roe, 2012)

