# OpenReview forum: "A Modern Take on the Bias-Variance Tradeoff in Neural Networks"
_ICLR.cc/2019/Conference_

### Official Review · AnonReviewer3 · 2018-11-03
**This paper needs to show some serious understanding on statistical machine learning**

**Rating:** 4
**Confidence:** 4

**Review:**

This paper suggests to rethink about the bias-variance tradeoff from statistical machine learning in the context of neural networks. Based on some empirical observations, the main claims in this work are that (1) it is not always the case that the variance will increase when we use bigger neural network models (particularly, by increasing the network width); (2) the variance should be decomposed into two parts: one part accounts for the variance caused by random initialization of network parameters/optimization and the other part is caused by "sampling of the training set".

For the first claim is based the empirical observation that increasing the number of hidden units did not cause the incrase of variance (as in figure 1). However, to my understanding, it only means increasing the number of hidden units is probably not a good way to increase the network capacity. In other words, this cannot be used as an evidence that the bias-variance tradeoff is not valid in neural network learning.

For the second claim, I don't like the way that they decompose the variance into two parts. To be clear, the classical bias-variance tradeoff doesn't consider the optimization error as an issue. For a more generic view of machine learning errors, please refer to "The Tradeoffs of Large Scale Learning" (Bottou and Bousquet, 2008). In addition, if the proposed framework wants to include the optimization error, it should also cover some other errors caused by optimization, for example, early stopping and the choice of a optimization algorithm.

Besides these high-level issues, I also found the technical parts of this paper is really hard to understand. For example,

- what is exactly the definition of $p(\theta|S)$? The closely related case I can think about is in the Baysian setting, where we want to give a prior distribution of model (parameter). But, clearly, this is not the case here.
- similar question to the "frequentist risk", in the definition of frequentist risk, model parameter $\theta$ should be fixed and the only expectation we need to compute is over data $S$
- in Eq. (5), I think I need more technical detail to understand this decomposition.

---

> ### Author Response · Authors · 2018-11-10
> **Author Response to Reviewer 3**
>
> Thank you for your feedback! It appears that there may have been an important miscommunication regarding our methodology; we hope our answers below will clarify this. Since clarity in the definitions in Section 2 is of utmost importance for understanding the paper, we’ve also added clarifications in our uploaded revision.
>
>
> On the two high-level points:
>
> 1. On width and capacity: If capacity refers to representation power, then increasingly large width networks have increasingly large capacity -- in fact  a wide enough network can fit any dataset [1]. The traditional view of bias-variance tradeoff is that increasingly large capacity models have lower bias and higher variance. This led Geman et al [2[ to claim that wide networks will suffer from high variance. We provide a quote of this claim and a sampling of related quotes from other impactful works in Appendix E. In our work we find that both bias and variance decrease with width, challenging the traditional view that bias and variance are related through a tradeoff merely governed by capacity.
>
> On effective capacity: We understand your comment as saying that this probably means that the very notion of capacity should be amended beyond simply representation power (e.g by explicitly taking into account optimization and the data). We completely agree with this. This is also the point made in Zhang et al. [4] and related work in the context of generalization gap analysis. We reach the same conclusion through a proper analysis of the variance of these models. To the best of our knowledge, this is new; this is the first study of the variance since [2] and it reaches opposite conclusions.
>
> 2. On our variance decomposition:  In contrast to the traditional bias-variance decomposition, which only considers one source of randomness (the training set), we are considering two sources of randomness: randomness from the optimization algorithm (mainly initialization) and randomness from sampling the training set. Going by the definition of optimization error in [3], we completely agree that the classical bias-variance tradeoff does not consider the optimization error. We reason that this is partially because variance due to initialization is 0 in the strongly convex case for a batch optimizer; given a decaying step size schedule, this is true for SGD as well [2, Section 4.2]. However, we are not trying to study the optimization error defined in [3]. We extended the classical bias-variance decomposition (via the law of total variance) to have another term that captures variance due to initialization because in the non-convex setting, the learned function is dependent on initialization. We found this extension yielded insightful results as the two variance terms have importantly different trends (Figure 2).
>
> As we mention in 3.2, our results in the full data setting were the same with or without early stopping. Also, as mentioned at the end of Section 3.3, we find the same trends with other optimization algorithms such as batch gradient descent and PyTorchs implementation of LBFGS (included in Appendix B.3).
>
>
> On the technical parts:
>
> Thank you for making it clear to us that the definitions in Section 2 were not given with sufficient precision; this feedback is very valuable to us. We have uploaded a revision that we hope will make this perfectly clear.
>
> 1. On $p( . |S)$: Given a training set $S$, the learned weight theta depends on the random initialization because of non-convexity. Hence it is not deterministic;  $p(\theta|S)$ is the distribution over the learned weights, conditioned on $S$. We have updated the corresponding  paragraph in Section 2, making explicit the random variable I that denotes initialization and explaining the relationship of the learning algorithm with S and I.
>
> 2. On frequentist risk:  it looks like there may be a misunderstanding with our notation. We use the standard notion of frequentist risk, just in a more general context. Unfortunately, the notation \theta usually refers to the population parameter, which may have caused some confusion. Our \theta denotes the learned weights of the neural network. Averaging over them w.r.t $p( . |S)$ amounts to averaging over initializations. Hopefully, our revisions in this section makes this all more clear.
>
> 3. On Eq 5: Hopefully our answers above clarify this equation. The variance is with respect to both initialization and data sampling. Eq. 5 then follows from the law of total variance. Please let us know if anything is still unclear.
>
>
> In closing:
> Thank you for your time. We hope you find that our responses and our revision address your concerns.
>
> [1] Neural Networks for Exact Matching of Functions on a Discrete Domain (Shrivastava and Dasgupta, 1990)
> [2] Understanding Deep Learning Requires Generalization (Zhang et al, 2017)
> [3] The Tradeoffs of Large Scale Learning (Bottou and Bousquet, 2008)
> [4] Optimization Methods for Large-Scale Machine Learning (Bottou et al., 2016)

---

> ### Author Response · Authors · 2018-12-01
> **Request for discussion/reconsideration**
>
> Thanks to your feedback, we have made the preliminaries more clear. We would appreciate if you'd consider these changes and the additional clarifications provided in our rebuttal. We hope you'll find that we've addressed your concerns and consider changing your score.

---

### Official Review · AnonReviewer1 · 2018-11-04
**Report on paper 1248**

**Rating:** 7
**Confidence:** 4

**Review:**

The paper offers a different and surprising view on the bias-variance decomposition. The paper shows, by a means of experimental studies and a simplified theoretical analysis, that variance decreases with the model complexity (in terms of the width of neural nets) , which is opposite to the traditional bias-variance trade-off.

While the conclusion is surprising, it is somewhat consistent with my own observation. However, there are potential confounding factors in such an experimental study that needs to be controlled for. One of these factors is the stability of the training algorithm being used. The variance term (and the bias) depends on the distribution p(theta|S) of the model parameters given data S. This would be the posterior distribution in Bayesian settings, but the paper considers the frequentist framework so this distribution encodes all the uncertainty due to initialisation, sampling and the nature of SGD optimizer being used. The paper accounts for the first two, but how about the stability of the optimiser? If the authors used a different optimizer for training, what would the variance behave then? A comment/discussion along this line would be interesting.

It is said in Section 3.1 that different random seeds are used for estimating both the outer and inter expectation in Eq. 5. Should the bootstrap be used instead for the outer expectation as this is w.r.t. the data? Another point that isn't clear to me is how the true conditional mean y_bar(x) = E(y|x)  is computed in real-data experiments, as this quantity is typically unknown.

---

> ### Author Response · Authors · 2018-11-10
> **Author Response to Reviewer 1**
>
> Thank you for the positive feedback!
>
> “stability of the training algorithm”:
> Thank you for bringing up this other factor to consider. There are 3 sources of randomness when using SGD (initialization, training set, and mini-batch sampling). We do not focus on variance due to mini-batch sampling because the the decreasing variance phenomenon persisted when using batch gradient descent (no randomness due to mini-batching); this result is included in Appendix B.3. In response to your feedback, we have added a footnote on the 3rd page that addresses this: “We do not study randomness from stochastic mini-batching because we found the phenomenon of decreasing variance with width persists when using batch gradient descent (Section 3.3, Appendix B.3).”
>
> Behavior of variance when using different optimizers:
> When we use other optimizers such as batch gradient descent and LBFGS, we still find that total variance decreases with width. These experiments are mentioned at the end of Section 3.3 and are included in Appendix B.3. We hope you find these results with other optimizers interesting.
>
> Random seeds for outer and inner expectations in Eq. 5:
> We greatly appreciate this comment. We believe there may be a misunderstanding, due to our writing in this section. The outer expectation is for estimation over randomness from training set sampling and the inner expectation is for estimation over randomness from initialization. This is necessary because the two terms from the law of total variance both depend on both sources of randomness; it’s just that they take variances with respect to different random variables. If only one seed were used for the inner expectation (randomness from initialization), we would be estimating a conditional variance (over training set sampling), which is conditioned on one specific initialization. To help make this more clear, we have added the explicit introduction of the random variable I, which denotes the randomness from optimization, in Section 2.1 and have added I to Eq. 5.
>
> “y_bar(x) = E(y|x)” and bias estimation:
> You are absolutely right that y_bar(x) = E(y|x) is unknown. We have added this clarification in footnote 4 of the new revision: “Because we don't have access to \bar{y}, we use the labels y to estimate bias. This is equivalent to assuming noiseless labels and is standard procedure for estimating bias (Kohavi and Wolpert, 1996; Domingos, 2000).”
>
> In closing:
> Thank you for your time. We hope that we have adequately addressed your questions and hope our revision makes the significance of main contribution 2 more apparent.

---

### Official Review · AnonReviewer2 · 2018-11-05
**Interesting paper with some experiments and preliminary results but requires more work**

**Rating:** 5
**Confidence:** 3

**Review:**

This paper studies variance-bias tradeoff as a function of depth and width of a neural network. Experiments suggest that variance may decrease as a function width and increase as a function of depth. Some analytical results are presented why this may the case for width and why the necessary assumptions for the depth are violated.

Main comment on experiments: if I am correct the step size for optimization is chosen in a data-dependent way for each size of the network. This is a subtle point since it leads to a data-dependent hypothesis set. In other words, in this experiments for each width we study variance of neural nets that can be found in fixed number of iterations by a step size that is chosen in data-dependent way. It may be the case that as width grows the step size decreases faster and hence hypothesis set shrinks and we observe decreasing variance. This makes the results of experiments with width not so surprising or interesting.

Further comments on experiments: it probably worth pointing out that results for depth are what we would expect from theory in general.

More on experiments: it would be also interesting to see how variance behaves as a function of width for depth other than 1.

On assumptions: it is not really clear why assumptions in 5.2 hold for wide shallow networks at least in some cases. Paper provides some references to prior work but it would be great to give more details. Furthermore, some statements seems to be contradicting: sentence before 5.2.1 seems to say that assumption (a) should hold for deep nets while sentence at the end of page 8 seems to say the opposite.

Overall: I think this paper presents an interesting avenue of research but due to aforementioned points is not ready for publication.

---

> ### Author Response · Authors · 2018-11-10
> **Author Response to Reviewer 2**
>
> Thank you for taking the time to review our paper!
>
>
> “Main comment on experiments [...] It may be the case that as width grows the step size decreases faster and hence hypothesis set shrinks and we observe decreasing variance”:
> Thank you for bringing up this point that leads to this natural hypothesis. Note that the bias is also going down with width. Traditional bias-variance tradeoff thinking associates decreasing bias with a growing hypothesis set. The fact that we see both bias and variance decrease is what’s surprising, as it shows we don’t need to trade bias for variance.
>
> In addition, we do not see that the step sizes are decreasing with width. The step sizes that were used for the decreasing variance in the small data setting (Figure 3a) are provided in Appendix B.1.
>
> Furthermore, note that the same experimental procedure did not lead to decreasing variance with depth. By the same line of reasoning as in the above quote, we would expect to get decreasing variance with depth by having smaller and smaller step sizes with deeper networks. However, our experimental procedure did not yield that.
>
> We hope that these points make it clear that we considered this potential explanation of our results, and we determined that this explanation does not capture the whole story. For more discussion on the justification of our experimental design, see Section 3.3 and Appendix B.2.
>
>
> “results for depth are what we would expect from theory in general”:
> While we agree that variance due to initialization is consistent with orthodoxy and discuss this in Section 4.2, we find the observation that variance due to sampling is roughly constant with depth (Figure 2b) quite surprising. This is because the traditional bias-variance tradeoff is exactly about training set sampling randomness (not optimization randomness). The distinction between these two sources of randomness is key to our deeper level of study (main contribution 2).
>
>
> On assumptions:
> We note that the assumptions are strong (though they have their basis in the referenced literature). Our primary goal is to give a rigorous argument beyond the linear case.
>
> Regarding your request for more details on the assumptions, we’ve updated the paragraph just before Section 5.2.1 with another reference and a more clear explanation: “Sagun et al. (2017) showed that the spectrum of the Hessian for over-parametrized networks splits into (i)  a bulk centered near zero and (ii)  a small number of large eigenvalues, which suggests that learning occurs mainly in a small number of directions.” This hypothesis was also formulated by Advani & Saxe (2017), and this is an active area of research. For example, there was a paper submitted to this ICLR titled “Gradient Descent Happens in a Tiny Subspace” that is entirely dedicated to this line of thinking: https://openreview.net/forum?id=ByeTHsAqtX  We view our analysis (and identification of these assumptions) as a useful contribution because it is consistent with the experimental results for width, and the assumptions have some basis in the literature.
>
> Additionally, we have added more to Section 5.3 to make the intuition more clear for why varying the depth is very different from varying the width, with respect to these assumptions.
>
>
> On seeming contradiction:
> Our reference to Advani & Saxe’s result was indeed very imprecise and seems to result in a contradiction. Thank you for pointing that out. We have updated this in the recent revision to report more clearly their result (just before Section 5.2.1).  Their analysis shows that our assumption (a) holds in deep linear networks under a simplifying assumption on the form of the weights that leads to a full decoupling of the dynamics of the weights at different layers. They claim this simplifying assumption is approximately true for small enough initial weights; and our empirical results suggest it might be increasingly inaccurate with depth.
>
>
> In closing:
> Thank you for your time. We hope you find that our revision addresses your concerns.

---

> > ### Comment · AnonReviewer2 · 2018-12-04
> > **thanks for clarifications**
> >
> > Thank you for clarifications. I think the following would make this more convincing:
> >
> > 1. As it is pointed out in the paper, it may be simply the case that you have not reached the bottom of the U curve in your experiments. Can you design a synthetic data experiment to clearly demonstrate that you should have reached that bottom?
> >
> > 2. This is an experimental paper (theory is rather limited), so ultimately I would expect to see experiments with a lot more datasets than just 2.5. If we observe the same phenomena with 10+ datasets that would be a lot more convincing.
> >
> > 3. Could you also add experiments with other models, e.g. trees, linear models that demonstrate that those are subject to the usual bias variance tradeoff? Again this is an experimental paper after all.
> >
> > 4. Ultimately, I am not convinced that it kosher to vary a hyperparameter (step size) between experiments. You diagram B.2. seems to exactly demostrate my concerns if I understand that correctly. In relation to that, it would be interesting to see how bias-variance curves look like for trees, linear models when we allow to vary their hyperparameters.
> >
> > I think as an experimental paper this would require more convincing experiments at the end of the day.

---

> > > ### Author Response · Authors · 2018-12-09
> > > **Author Response to Additional Feedback**
> > >
> > > Thank you for the additional feedback.
> > >
> > > 1. We can only run experiments with as large networks as our hardware allows. The point of the small data experiment is to provide further evidence that the total variance continues to monotonically decreases up to very large widths (relative to the size of the dataset). We take the fact that the variance plateaus for so long (with network width on the x-axis) as evidence that it will not eventually increase (forming a U shape). An experiment with any type of data would have to stop at some network size and take the long plateau as evidence that the variance will not go up.
> > >
> > > 2. While we agree that more datasets is always better, these experiments are quite compute intensive. We developed the theory in Section 5 (Eqs 9-11 and Theorem 1) to better illustrate why this phenomenon is happening.
> > >
> > > 3. The focus of this paper is neural networks. Traditional bias-variance tradeoff curves on other models such as KNNs and kernel regression [1] and trees and boosting [2] have been demonstrated previously. In textbooks, there are more examples of traditional bias-variance tradeoffs on other models (see e.g. Section 3.2 of [3], Section 6.4.4 of [4], etc.). In over-parameterized linear models, we actually do not expect traditional bias-variance tradeoff curves, based on what we present in Equations 9-11.
> > >
> > > 4. First, note that we observe this decreasing total variance trend in the full data setting when all networks use the same step size (Figure 1b). In the small data setting, we use a validation set to choose the step size for each different architecture, which, we acknowledge, is a form of regularization (footnote 3). Most importantly, we demonstrate that BOTH bias and variance are decreasing with capacity, indicating that it is not necessary to trade bias for variance. Regarding the appropriateness of choosing this single hyperparameter (step size) this way, it is known that hyperparameter tuning is important for comparing different models fairly as the same hyperparameters (e.g. step size) do not often transfer across models. The figure in Appendix B.2 is a consequence of the fact that hyperparameter settings often do not transfer across architectures.
> > >
> > > [1] Neural Networks and the Bias/Variance Dilemma (Geman et al., 1992)
> > > [2] A Unified Bias-Variance Decomposition and its Applications (Domingos, 2000)
> > > [3] Pattern Recognition and Machine Learning (Bishop, 2006)
> > > [4] Machine Learning: a Probabilistic Perspective (Murphy, 2013)

---

> > > > ### Comment · AnonReviewer2 · 2018-12-09
> > > > **reply**
> > > >
> > > > Thank you for your reply. Unfortunately, without more convincing experiments (along the lines suggested in my previous comment), this work appears incomplete.
> > > >
> > > > I appreciate the fact that you may not have the necessary compute resources to carry out these experiments but that does not justify publication of this work.
> > > >
> > > > I also disagree that if the focus is on NNs then experiments with other models are not needed. Trying other models is effectively a baseline in this case.

---

> > > > > ### Author Response · Authors · 2018-12-09
> > > > > **Thank you for your time**
> > > > >
> > > > > Thank you for your time.

---

> ### Author Response · Authors · 2018-12-01
> **Request for discussion/consideration**
>
> Thanks to your feedback, we have made improvements to our paper. We would appreciate if you'd consider these changes and the additional clarifications provided in our rebuttal. We hope you'll find that we've addressed your concerns and consider changing your score.

---

### Author Response · Authors · 2018-11-10
**Global Comment from Authors**

Thank you to all of the reviewers for their time. Most importantly, we have revised Section 2.1 to make it more clear and have added an explicit description of all randomness in Eq. 5 in order to aid in the understanding of our decomposition of variance (main contribution 2). We hope this improved clarity in the preliminaries makes the significance of main contribution 2 more apparent.

Main contributions:
1. Variance decreases with width (along with bias), indicating that it isn’t necessary to trade bias for variance.
2. We perform a deeper study of variance by decomposing the coarse variance into two terms: variance due to training set sampling (like the classical decomposition) and variance due to initialization. We find variance due to sampling is roughly constant with both width and depth (Figure 2).
3. In a simplified setting, inspired by linear models, we provide theoretical analysis in support of our empirical findings for network width.

---

### Meta-Review · Area_Chair1 · 2018-12-14

**Confidence:** 4
**Recommendation:** Reject

**Metareview:**

The paper revisits the traditional bias-variance trade-off for the case
of large capacity neural networks. Reviewers requested several clarifications
on the experimental setting and underlying results. Authors provided some,
but it was deemed not enough for the paper to be strong enough to be accepted.
Reviewers discussed among themselved but think that given the paper is mostly
experimental, it needs more experimental evidence to be acceptable.
Overall, I found the paper borderline but concur with the reviewers to reject
it in its current form.